# FusionBench: A Comprehensive Benchmark of Deep Model Fusion

## Abstract

Deep model fusion is an emerging technique that unifies the predictions or parameters of several deep neural networks into a single model in a cost-effective and data-efficient manner. This enables the unified model to take advantage of the original models' strengths, potentially exceeding their performance. Although a variety of deep model fusion techniques have been introduced, their evaluations tend to be inconsistent and often inadequate to validate their effectiveness and robustness against distribution shifts. To address this issue, we introduce *FusionBench*, which is the first comprehensive benchmark dedicated to deep model fusion. FusionBench covers a wide range of tasks, including open-vocabulary image classification, text classification, and text-to-text generation. Each category includes up to eight tasks with corresponding task-specific models, featuring both full fine-tuning and LoRA fine-tuning, as well as models of different sizes, to ensure fair and balanced comparisons of various multi-task model fusion techniques across different tasks, model scales, and fine-tuning strategies. We implement and evaluate a broad spectrum of deep model fusion techniques. These techniques range from model ensemble methods, which combine the predictions to improve the overall performance, to model merging, which integrates different models into a single one, and model mixing methods, which upscale or recombine the components of the original models. FusionBench now contains a range of CV and NLP tasks, 74 fine-tuned models, and 19 fusion techniques, and we are committed to consistently expanding the benchmark with more tasks, models, and fusion techniques. In addition, we offer a well-documented set of resources and guidelines to aid researchers in understanding and replicating the benchmark results. This includes detailed documentation, code examples, and tutorials, making FusionBench a user-friendly and accessible platform for both beginners and experienced researchers.

## 1 Introduction

In recent years, a new paradigm called "learn from model" has emerged in the field of deep learning, which focuses on leveraging the knowledge embedded in existing models to develop new ones (Zheng et al., 2023). This paradigm has been widely adopted in various scenarios, such as model tuning (He et al., 2022; Chung et al., 2024), model distillation (Hinton, 2015), model pruning (Han et al., 2015; Asif et al., 2020), model editing (Mitchell et al., 2021; Zhang et al., 2024), and so on. Among these methods, deep model fusion is particularly appealing. It merges the parameters or predictions of multiple models to create a more robust and efficient unified model. Due to its effectiveness and scalability, many new techniques for deep model fusion have recently been proposed (Li et al., 2023).

Deep model fusion offers both scalability and data efficiency by utilizing the knowledge embedded in pre-existing models, rather than requiring training from scratch. This approach significantly accelerates model development, making it a practical solution in the current era dominated by large foundation models. Despite its potential, the evaluation of deep model fusion techniques often suffers from inconsistency and inadequacy. Standardized assessments are lacking, making it challenging to verify their effectiveness and robustness. The potential reasons for this inconsistency include the rapid development of new techniques, the absence of standardized tasks and models, and the variety of settings (such as different fine-tuning strategies). Additionally, challenges in implementing or replicating prior work contribute to these inconsistencies.

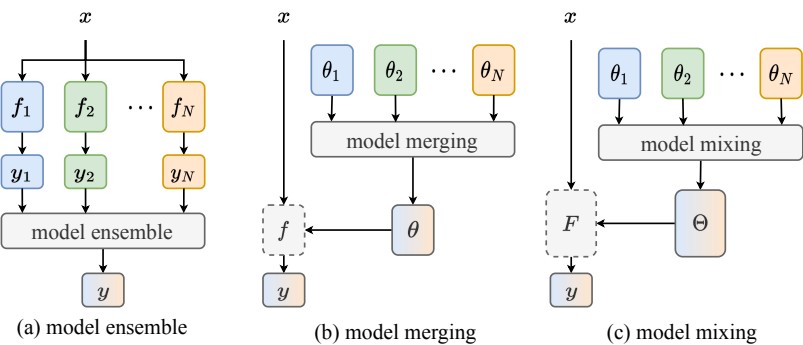

Figure 1: A taxonomy of deep model fusion techniques.

To tackle these challenges, we develop the first comprehensive benchmark dedicated to deep model fusion, called **FusionBench**. FusionBench is built to be a modular and extensible platform, comprising three core modules: the *Algorithm Module*, the *Model Pool Module*, and the *Task Pool Module*. Each module is configurable, allowing users to easily customize and manage the benchmark settings. In summary, our main contributions are four-fold:

1. **An extensive codebase and configurable interface:** Our codebase is structured around three core modules: the algorithm module, the model pool module, and the task pool module. Moreover, to facilitate ease of use and flexibility, we provide a unified command line interface with YAML configuration options for effortless customization.
2. **Comprehensive evaluations:** We conduct thorough evaluations of the deep model fusion techniques, assessing their performance across a variety of benchmarks and settings.
3. **Extensive analysis and insights:** We provide detailed analysis and insights based on the evaluation results, revealing key factors that influence their effectiveness. This includes identifying best practices, exploring the impact of fine-tuning strategies, and offering recommendations for future research.
4. **User-friendly resources and guidelines:** We offer a well-documented set of resources and guidelines to aid researchers in understanding and replicating the benchmark results. This includes detailed documentation, code examples, and tutorials, making FusionBench a user-friendly and accessible platform for both beginners and experienced researchers.

## 2 RELATED WORK

Since deep model fusion is a relatively new research area, there is currently no standardized taxonomy. Different researchers may categorize these techniques in various ways based on their understanding and points of view. Here, we propose a taxonomy that divides these techniques into three major categories: *Model Ensemble*, *Model Merging*, and *Model Mixing*. Each of these categories approaches model fusion from a unique perspective, offering distinct advantages and applicability. In the following, we provide detailed explanations, formal definitions, and analyze their strengths and weaknesses. A visualization of the taxonomy is shown in Figure 1.

**Model Ensemble** methods combine the predictions of multiple models to improve the overall performance of a machine learning system (Sagi & Rokach, 2018), where the collective knowledge is often more accurate and reliable than that of any individual model. Mathematically, given a set of $N$ models $\{f_1, f_2, \ldots, f_N\}$, which can be homogeneous or heterogeneous, we use their predictions to obtain a global prediction $y = \mathcal{A}_{ensemble}(x; f_1, f_2, \ldots, f_N; w)$, where $\mathcal{A}_{ensemble}$ is an ensemble algorithm and $w$ are the algorithmic parameters. Each model $f_i$ can also be associated with a specification to indicate its weight or importance in the ensemble (Pathak et al., 2010; Zhou, 2016; Wu et al., 2021; Tang et al., 2023a). Ensemble methods are widely used and effective in improving performance but are often expensive to use and manage. Recent research has also investigated efficient techniques for model ensembles (Wen et al., 2020; Chen et al., 2023; Allingham et al., 2021).

**Model Merging** methods integrate the parameters of multiple models into a unified model, enhancing efficiency in terms of inference cost and storage, and enabling scalable model fusion. Given a set of

$N$ isomorphic models $\{f_i(\cdot; \theta_i)\}_{i=1}^{N}$, each parameterized with $\theta_i$, we merge them into a single model with parameters $\theta = \mathcal{A}_{merging}(\theta_1, \theta_2, \ldots, \theta_N; w)$, where $\mathcal{A}_{merging} : \mathbb{R}^{N \times d} \to \mathbb{R}^d$ is a merging algorithm and $w$ are the algorithmic parameters. The merged model can be expressed as $f(\cdot; \theta)$. This method can be implemented through linear interpolation in parameter space (Wortsman et al., 2022; Ilharco et al., 2022; Yadav et al., 2023; Matena & Raffel, 2022; Yu et al., 2024; Chronopoulou et al., 2023; Rame et al., 2024; Ortiz-Jimenez et al., 2024; Liu & Soatto, 2023), leveraging mode connectivity (Draxler et al., 2018; Frankle et al., 2020; Benton et al., 2021; Garipov et al., 2018; Qu & Horvath, 2024), aligning features, parameters or gradients (Liu et al., 2022; Ainsworth et al., 2022; Jin et al., 2022; Tam et al., 2024; Stoica et al., 2023; Jang et al., 2023; Daheim et al., 2023; Yang et al., 2024), subspace-based methods (Tang et al., 2023b; Wang et al., 2024; Yi et al., 2024; Zhu et al., 2024; Xu et al., 2024), and ensemble distillation (Wan et al., 2024a;b). Model merging methods are often performed in a data-efficient manner, the algorithmic parameters $w$ can also be learned during test time via test-time adaptation (TTA) training or meta-learning for a more seamless merging (Yang et al., 2023; Tang et al., 2023b).

**Model Mixing** methods fuse the components of multiple models to create a new heterogeneous model, which can be more flexible and adaptive than the original models. Mathematically, given a set of $N$ models $\{f_i(\cdot; \theta_i)\}_{i=1}^{N}$, each parameterized with $\theta_i \in \mathbb{R}^d$, we mix their components to obtain a new model with parameters $\Theta = \mathcal{A}_{mixing}(\theta_1, \theta_2, \ldots, \theta_N; w) \in \mathbb{R}^{d'}$, where $\mathcal{A}_{mixing} : \mathbb{R}^{N \times d} \mapsto \mathbb{R}^{d'}$ is a mixing algorithm and $w$ is the algorithmic parameters. The mixed model can be expressed as $F(\cdot; \Theta)$, which often has more parameters than the original models, and thus can be more expressive and powerful to capture the underlying patterns in the data. Model mixing methods can be implemented through layer recombinations (Hu et al., 2023; Jiang, 2024), model stitching (Lenc & Vedaldi, 2015; Moschella et al., 2022), or upscale to create a Mixture of Experts (MoE)-based sparse model (Komatsuzaki et al., 2022; Ye & Xu, 2023; Tang et al., 2024c; Lu et al., 2024; Dai et al., 2024; Zhao et al., 2024; Ostapenko et al., 2024; Tang et al., 2024b; Yadav et al., 2024).

Although several model fusion methods have been proposed, benchmarks and unified toolkits are still lacking in this field. A recent notable work, MergeKit (Goddard et al., 2024), provides a collection of model fusion techniques *specifically designed for merging large language models (LLMs)*, with a focus on model merging methods and Transformer-based LLMs. However, MergeKit's scope is limited to a specific domain and model architecture, while FusionBench is more comprehensive and covers a wider range of deep model fusion algorithms, as well as tools for evaluating these algorithms. *In general, FusionBench is more research-oriented.* It includes a diverse set of fine-tuned models and tasks to evaluate, making it a more generalized and versatile platform for assessing the performance of different model fusion approaches across various domains and architectures.

## 3 OUR BENCHMARK

The general framework of the modularized FusionBench codebase is shown in Figure 2, which consists of three primary elements: *Algorithm Module*, *Model Pool Module*, and *Task Pool Module*. In Section 3.1, we introduce the codebase, which is designed to be flexible and modular, allowing users to easily run experiments and evaluate the performance of model fusion algorithms. In Section 3.2 and Section 3.3, we introduce the implemented model fusion algorithms and the tasks and models included in FusionBench. Finally, in Section 3.4, we discuss the documentation and tutorials provided to help users understand the benchmark and effectively use the codebase. In Appendix A, we provide a flowchart to illustrate the process of running experiments and evaluating the merged models.

### 3.1 CODEBASE

We've constructed a flexible and modular codebase, which serves as the foundation for *FusionBench*. As shown in Figure 2, the codebase is composed of three primary elements: *Algorithm Module*, *Model Pool Module*, and *Task Pool Module*, which are responsible for implementing the model fusion algorithms, managing the models to be fused, and managing the tasks to be evaluated, respectively. Additionally, we provide a command line interface (CLI) to facilitate the use of the codebase and to enable users to easily run experiments and evaluate the performance of model fusion algorithms.

- **Algorithm Module** is the core component of the codebase, which contains the implementation of various model fusion algorithms. Each algorithm is implemented as a separate Python class, which

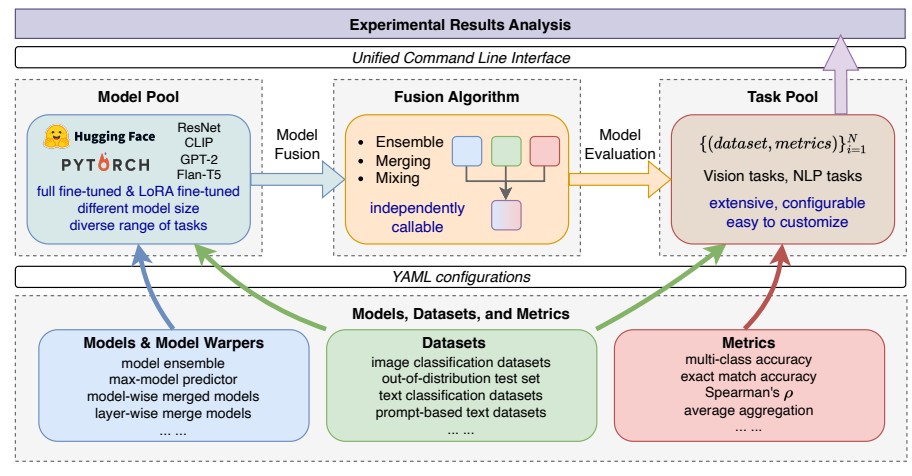

Figure 2: The general framework of the modularized FusionBench codebase.

Table 1: Implemented model fusion algorithms in FusionBench.

| CTGY. | METHOD | REQUIREMENT |
|---|---|---|
| Ensemble | Simple Ensemble (Sagi & Rokach, 2018) | - |
| | Weighted Ensemble (Sagi & Rokach, 2018) | hyperparameter search |
| | Max-Model Predictor (Wu et al., 2019) | - |
| Merging | Simple Average / Modelsoups (Wortsman et al., 2022) | - |
| | Weighted Average (Matena & Raffel, 2022) | hyperparameter search |
| | Fisher Merging (Matena & Raffel, 2022) | compute weights on labeled data |
| | RegMean (Jin et al., 2022) | compute weights on labeled data |
| | Task Arithmetic (Ilharco et al., 2022) | hyperparameter search |
| | Ties-Merging (Yadav et al., 2023) | hyperparameter search |
| | Task-Wise AdaMerging (Yang et al., 2023) | test-time adaptation training |
| | Layer-Wise AdaMerging (Yang et al., 2023) | test-time adaptation training |
| | Concrete Subspace (Tang et al., 2023b) | test-time adaptation training |
| Mixing | Depth Upscaling (Kim et al., 2023) | pre-training to recover performance |
| | MoE-based Upscaling (Komatsuzaki et al., 2022) | pre-training to recover performance |
| | MoE-based Merging (Komatsuzaki et al., 2022) | training on the combined model |
| | Weight-Ensemble MoE (Tang et al., 2024c) | test-time adaptation training, vision tasks |
| | Pareto-Driven Merging (Tang et al., 2024a) | training datasets |
| | SMILE Upscaling (Tang et al., 2024b) | - |
| | Model Recombination (Hu et al., 2023) | training on the combined model |

inherits from the base class `ModelFusionAlgorithm`. The algorithm classes are designed to be configurable and independently callable, allowing users to easily instantiate and set up the algorithms through our CLI or by directly invoking the Python classes in their own code.

- **Model Pool Module** is responsible for managing the models to be fused. It offers a unified interface for loading the pre-trained model and fine-tuned models. The module is designed to be extensible, allowing users to easily add support for new model architectures and add their own models to the pool. Each model in the pool can also be associated with metadata to meet the requirements of specific model fusion algorithms, such as the test dataset for test-time adaptation training.

- **Task Pool Module** is responsible for managing the tasks to be evaluated. Each task comprises a dataset and a set of evaluation metrics, which are defined in the YAML configuration file. This module offers a unified interface for loading tasks and assessing the performance of model fusion algorithms on these tasks. Users can effortlessly add support for new task types and evaluation metrics, or add new tasks of the same type but with different datasets.

Table 2: Tasks and models included in FusionBench for evaluating multi-task model fusion algorithms.

| DOMAIN | TASK TYPE | DATASETS | MODELS |
|---|---|---|---|
| Computer Vision | Image classification (8 domains) | SUN397, Stanford Cars (Krause et al., 2013), RESISC45 (Cheng et al., 2017), EuroSAT (Helber et al., 2018), SVHN (Netzer et al., 2011), GTSRB (Stallkamp et al., 2012), MNIST (Lecun et al., 1998), DTD (Cimpoi et al., 2014) | 8×CLIP-ViT-B/32, 24×CLIP-ViT-B/16 (w/ LoRA, L-LoRA), 8×CLIP-ViT-L/14 |
| | Sence Understanding (3 tasks) | NYUv2 (Silberman et al., 2012) | 3×Resnet-50 models |
| Natural Language Processing | Text classification (7 domains) | CoLA, MNLI, MRPC, QNLI, QQP, RTE, and SST-2 (Wang, 2018) | 7×GPT-2 |
| | Text-to-text generation (8 tasks) | CoLA, MNLI, MRPC, QNLI, QQP, RTE, SST-2, and STSB (Wang, 2018) | 16×Flan-T5-Base (w/ & w/o LoRA), 8×Flan-T5-Large (w/ LoRA) |

(a) Clean    (b) Motion    (c) Impulse    (d) Gaussian    (e) Pixelate    (f) Spatter    (g) Contrast    (h) JPEG

Figure 3: Here are eight instances of distorted images from the Stanford Cars dataset, which are used to assess the robustness and generalization capacity of the TTA-based merging algorithms.

## 3.2 IMPLEMENTED ALGORITHMS

In our benchmark, we have implemented 16 model fusion algorithms as the initial set. This includes 3 model ensemble methods, 8 model merging methods, and 5 model mixing methods. Our primary selection criterion for choosing among various algorithms is their applicability and effectiveness within the realm of deep learning architectures We have also considered the popularity of the algorithms in the literature and their practical applicability, such as their potential use in large-scale language models. We list the implemented algorithms in Table 1.

As shown in Table 1, we implemented three kinds of model fusion algorithms. A brief introduction and formal definition of our taxonomy are provided in Section 2. Model ensemble methods are effective at enhancing the performance of a machine learning system, but they are computationally expensive to infer. Model merging methods aim to integrate the advantages of individual models, making them popular in multi-task model fusion and auxiliary learning. In these scenarios, multiple single-task models are merged to construct a multi-task model, or models focused on auxiliary tasks are combined to boost the performance of a primary task. Model mixing methods are frequently used to scale up a pre-trained model to a larger size or to combine multiple models into a new one. Consequently, model mixing methods often necessitate additional training after the fusion process.

## 3.3 TASKS AND MODELS

Model fusion is a versatile technique that can be applied across various machine learning tasks at different stages of model development. In FusionBench, we specifically provide a diverse array of tasks and corresponding fine-tuned models to ensure a fair and comprehensive evaluation of **multi-task model fusion algorithms**. We have selected tasks from the domains of computer vision and natural language processing, as these are the most popular and extensively studied areas in deep

learning research. The tasks included in our benchmark are open-vocabulary image classification, text classification, and text-to-text generation. We list these tasks and models in Table 2. We make them publicly available to facilitate reproducibility and further research at HuggingFace.

- **Open-vocabulary image classification** is a challenging task that requires models to classify images into a large number of categories. We have selected eight image classification datasets, including SUN397 (Xiao et al., 2010), Stanford Cars (Krause et al., 2013), RESISC45 (Cheng et al., 2017), EuroSAT (Helber et al., 2018), SVHN (Netzer et al., 2011), GTSRB (Stallkamp et al., 2012), MNIST (Lecun et al., 1998), DTD (Cimpoi et al., 2014). These datasets cover a wide range of image classification tasks, including object recognition, satellite image classification, and texture classification. Customized tasks can be easily added to the benchmark by configuring the YAML file. We fine-tuned two CLIP-ViT models, CLIP-ViT-B/32 and CLIP-ViT-L/14, on these datasets. We report accuracy as the evaluation metric for these tasks. Specifically, to assess the robustness of multi-task model fusion algorithms, particularly those needing test-time adaptation training, we adopt the techniques recommended by Hendrycks & Dietterich (2019) to create corrupted versions of the test set for Cars, EuroSAT, RESISC45, and GTSRB. These corruptions are designed to simulate common image corruptions in real-world scenarios, including motion blur, impulse noise, Gaussian noise, pixelation, spatter, contrast adjustments, and JPEG compression.

- **Scene understanding tasks** are performed using the NYUv2 (Silberman et al., 2012) dataset, which consists of RGB-D images and includes three tasks: 13-class segmentation, depth estimation, and surface normal estimation. We fine-tuned ResNet-50 models (He et al., 2016) as the backbone for our experiments. The initial weights for these models were pre-trained on the ImageNet dataset. We then adapted them to the specific downstream tasks.

- **Text classification** is a fundamental task in natural language processing that involves categorizing text data into predefined classes. We have selected seven text classification tasks from the General Language Understanding Evaluation (GLUE) benchmark (Wang, 2018), including CoLA, MNLI, MRPC, QNLI, QQP, RTE, and SST-2. We fine-tuned GPT-2 models on these seven tasks, each with a different head for classification (Radford, 2018). We report accuracy as the evaluation metric.

- **Text-to-text generation** poses greater challenges compared to text classification, as it necessitates generating appropriate text outputs rather than mapping hidden representations to logits. Similar to text classification, we have selected eight text-to-text generation tasks from the GLUE benchmark, including CoLA, MNLI, MRPC, QNLI, QQP, RTE, SST-2, and STSB. We fine-tuned Flan-T5 models on these tasks, with and without the LoRA adaptation (Hu et al., 2021). The prompt templates for these tasks are provided in Appendix E. As for the evaluation metric, we report Spearman's $\rho$ for STSB and exact match accuracy for other tasks.

### 3.4 Documentation, Tutorials, and Theoretical Framework of Model Fusion

Documentation and tutorials are essential for beginners to understand the methodology behind the benchmark, to reproduce the experiments, and to effectively use the codebase. To this end, we offer comprehensive documentation and tutorials on the project homepage, which guide users through the fundamentals of model fusion, the steps to run experiments, and the procedures for evaluating the performance of model fusion algorithms. Additionally, we present some experimental results to shed light on the performance of different model fusion algorithms across various tasks.

As for the theoretical framework and insight of model fusion, each category of fusion algorithms operates on distinct theoretical foundations and assumptions, making it challenging to provide a comprehensive overview within the confines of a single paper. To illustrate: (1) Ensemble methods are rooted in the "wisdom of the crowd" principle (Sagi & Rokach, 2018). This approach posits that combining multiple models can yield superior performance compared to any individual model; (2) Weight interpolation-based model merging methods typically typically based on the findings of linear mode connectivity in deep neural networks, i.e. The existence of linear paths of low loss between solutions of optimization (Freeman & Bruna, 2016; Simsek et al., 2021); (3) Mixing methods, such as MoE-based model upscaling methods (Yadav et al., 2024), are founded on the understanding that *parameter/task interference* is a prevalent issue in multi-task model fusion (Yu et al., 2024; Tang et al., 2024b; Wang et al., 2024). These methods recognize that this interference problem is difficult to be effectively addressed within the original weight space. This diversity in theoretical underpinnings highlights the complexity and richness of the model fusion landscape. We provide a suggested reading list along with the documentation to help users delve deeper into these topics.

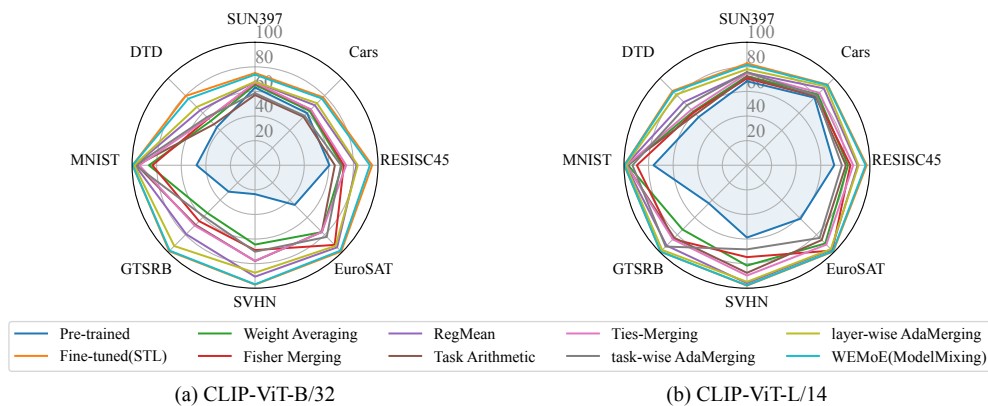

(a) CLIP-ViT-B/32        (b) CLIP-ViT-L/14

Figure 4: Radar charts comparing the performance of different model fusion methods across multiple tasks using CLIP-ViT-B/32 and CLIP-ViT-L/14.

Table 3: Multi-task performance when merging CLIP-ViT-B/32 models on all eight tasks.

| METHOD | SUN397 | Cars | RESISC45 | EuroSAT | SVHN | GTSRB | MNIST | DTD | AVG. |
|---|---|---|---|---|---|---|---|---|---|
| *Reference Methods* | | | | | | | | | |
| Pre-trained | 63.2 | 59.8 | 60.7 | 46.0 | 31.6 | 32.5 | 48.2 | 43.9 | 48.2 |
| Individual Fine-tuned | 75.0 | 78.3 | 95.2 | 99.0 | 97.3 | 98.9 | 99.6 | 79.7 | 90.3 |
| Traditional MTL | 72.3 | 76.6 | 92.2 | 97.9 | 95.5 | 97.7 | 99.3 | 77.7 | 88.6 |
| *Multi-Task Model Fusion Methods* | | | | | | | | | |
| Weight Averaging | 65.4 | 62.6 | 70.8 | 76.9 | 64.5 | 54.9 | 86.3 | 50.9 | 66.5 |
| Fisher Merging | 66.7 | 64.0 | 72.2 | 91.6 | 69.0 | 64.3 | 83.5 | 53.7 | 70.6 |
| RegMean | 67.8 | 68.9 | 82.5 | 94.4 | 90.6 | 79.2 | 97.6 | 63.2 | 80.5 |
| Task Arithmetic | 57.1 | 55.7 | 64.9 | 76.7 | 77.9 | 68.5 | 96.1 | 47.2 | 68.0 |
| Ties-Merging | 67.1 | 64.2 | 74.1 | 76.8 | 77.7 | 69.4 | 94.1 | 54.0 | 72.2 |
| task-wise AdaMerging | 58.6 | 56.9 | 69.8 | 82.4 | 70.3 | 58.9 | 97.2 | 55.3 | 68.7 |
| layer-wise AdaMerging | 67.9 | 71.3 | 83.5 | 92.7 | 87.4 | 92.9 | 98.2 | 67.0 | 82.6 |
| WEMoE (Model Mixing) | 73.7 | 76.8 | 93.4 | 98.2 | 96.8 | 98.2 | 99.6 | 76.6 | 89.2 |
| SMILE (Model Mixing) | 73.6 | 77.8 | 92.0 | 98.3 | 96.9 | 98.1 | 99.6 | 78.1 | 89.3 |

## 4 EVALUATION AND ANALYSIS

In this section, we evaluate the performance of multi-task model fusion algorithms on a variety of tasks, as well as analyze the generalization and robustness of these algorithms. We also provide an ablation study to investigate the impact of hyperparameter selection. Most of the experiments are conducted with a single NVIDIA RTX 3090 GPU with 24GB memory.

### 4.1 EXPERIMENTAL SETUP

In this section, we conduct a series of multi-task model fusion experiments on image classification tasks, text classification tasks, and text-to-text generation tasks to evaluate the performance of multi-task model fusion algorithms. These tasks are chosen to cover a wide range of NLP and CV tasks, as described in Section 3.3. Table 2 provides a summary of the tasks and models used in our experiments.

### 4.2 MULTI-TASK MODEL FUSION

In this evaluation, we begin by comparing multi-task model fusion algorithms under several settings:

1. **Image Classification**: We use CLIP models from HuggingFace. Results for CLIP-ViT-B/32 are in Table 3 and Figure 4(a), and for CLIP-ViT-L/14 in Table 15 and Figure 4(b).

Table 4: Experimental results of merging single-task Resnet50 models on three NYUv2 tasks.

| METHOD | SEGMENTATION | | DEPTH ESTIMATION | | NORMAL |
| | mIoU ↑ | Pix Acc ↑ | Abs Err ↓ | Rel Err ↓ | Mean ↓ |
|---|---|---|---|---|---|
| *Single-Task Learning* | | | | | |
| Segmentation | **52.0** | **73.8** | 242.8 | 88.7 | 82.8 |
| Depth Estimation | 2.3 | 6.2 | **42.5** | **17.7** | 82.8 |
| Normal | 2.0 | 4.9 | 264.0 | 98.1 | **24.7** |
| *Multi-Task Model Fusion Methods* | | | | | |
| Weight Averaging | 39.0 | 67.0 | 55.1 | 22.7 | 30.4 |
| Task Arithmetic ($\lambda = 0.3$) | 33.6 | 63.3 | 56.3 | 23.2 | 31.3 |
| Ties-Merging ($\lambda = 0.3$) | 36.3 | 61.7 | 60.5 | 24.5 | 33.1 |

Table 5: Generalization results on two unseen tasks when merging ViT-B/32 models on six tasks.

| METHOD | Seen Tasks (ACC) | | | | | | | Unseen Tasks (ACC) | | |
| | SUN397 | Cars | RESISC45 | DTD | SVHN | GTSRB | Avg. | MNIST | EuroSAT | Avg. |
|---|---|---|---|---|---|---|---|---|---|---|
| Pre-trained | 63.2 | 59.9 | 60.6 | 43.9 | 23.5 | 30.4 | 46.9 | 47.6 | 45.6 | 46.6 |
| Fisher Merging | 65.5 | 67.2 | 78.2 | 57.6 | 84.2 | 75.9 | 71.4 | 71.8 | 49.4 | 60.6 |
| RegMean | 68.7 | 70.0 | 86.5 | 65.9 | 93.9 | 86.7 | 78.6 | 82.2 | 49.3 | 65.7 |
| Task Arithmetic | 64.3 | 63.0 | 73.2 | 54.9 | 84.7 | 79.5 | 69.9 | 75.5 | 42.6 | 59.1 |
| Ties-Merging | 68.3 | 65.5 | 76.9 | 54.9 | 75.4 | 72.0 | 68.9 | 73.1 | 47.3 | 60.2 |
| AdaMerging | 68.4 | 71.9 | 87.9 | 69.1 | 92.2 | 93.8 | 80.5 | 77.7 | 47.3 | 62.5 |
| WEMoE | 75.4 | 77.5 | 94.3 | 77.0 | 96.8 | 98.7 | 86.6 | 78.3 | 44.0 | 61.1 |

2. **Scene Understanding**: Using ResNet-50 models on the NYUv2 dataset for segmentation, depth estimation, and normal estimation tasks. Results are in Table 4.
3. **Text Classification**: Results for GPT-2 models on seven tasks are shown in Table 13.
4. **Text-to-Text Generation**: For LoRA fine-tuned Flan-T5-base and Flan-T5-large models, after merging and unloading the LoRA adapters, results are in Tables 14 and 16.

In these tables, we compare the performance of different multi-task model fusion algorithms across various tasks. Pre-trained models' performance, fine-tuned models' performance, and traditional multi-task learning (MTL) methods are provided for reference. In Appendix B, we provide a detailed description of these fine-tuned single-task models.

We have the following key observations: (1) Multi-task model fusion usually outperforms pre-trained models, showing it can transfer knowledge from multiple single-task models to enhance performance. Pre-trained models lack task-specific knowledge as they are not fine-tuned for downstream tasks. (2) Adaptive method (AdaMerging) and MoE-based method perform best in multi-task model fusion, showing the effectiveness of adaptive merging and mixture-of-experts approaches. (3) The performance gap between multi-task model fusion and single-task fine-tuned models (STL) is larger for CLIP-ViT-B/32 compared to CLIP-ViT-L/14. This suggests that multi-task model fusion may be more beneficial for smaller models, as they have more room for improvement through knowledge transfer. (4) Traditional MTL outperforms most multi-task model fusion methods, which indicates that traditional MTL is still a strong baseline for multi-task learning, and there is room for improvement in multi-task model fusion algorithms.

### 4.3 GENERALIZATION AND ROBUSTNESS EVALUATION

To further assess the generalization and robustness of multi-task model fusion algorithms, we conduct experiments on *unseen tasks* and *corrupted test sets* (or *out-of-distribution test sets*). (1) Tables 5 and 17 present the generalization performance of various multi-task model fusion algorithms when merging CLIP-ViT-B/32 models trained on six seen tasks and evaluating their performance on two unseen tasks. This analysis helps us understand how well the fused models can adapt to new tasks

Table 6: Ablations of the test data distribution on ViT-B/32 (for all methods, $\lambda = 0.3$).

| METHOD | Cars | EuroSAT | RESISC45 | GTSRB | Avg. | Cars | EuroSAT | RESISC45 | GTSRB | Avg. |
|---|---|---|---|---|---|---|---|---|---|---|
| | Clean Test Set | | | | | Corrupted Test Set (Motion Blur) | | | | |
| Fisher Merging | 66.0 | 92.7 | 83.7 | 78.7 | 80.3 | 60.7 | 57.6 | 81.7 | 78.4 | 69.6 |
| RegMean | 72.1 | 97.5 | 88.9 | 93.9 | 88.1 | 70.0 | 71.3 | 87.5 | 86.8 | 78.9 |
| Task Arithmetic | 64.6 | 91.8 | 80.2 | 74.8 | 77.9 | 62.4 | 59.2 | 78.5 | 63.3 | 65.9 |
| Ties-Merging | 65.2 | 83.3 | 78.1 | 67.4 | 73.5 | 64.4 | 53.9 | 76.4 | 57.1 | 62.9 |
| AdaMerging | 75.2 | 94.3 | 87.6 | 96.7 | 88.5 | 72.4 | 72.7 | 85.3 | 94.3 | 81.2 |
| WEMoE | 77.4 | 98.9 | 94.4 | 99.0 | 92.4 | 76.5 | 74.2 | 93.7 | 97.4 | 85.5 |
| | Corrupted Test Set (Impluse Noise) | | | | | Corrupted Test Set (Gaussian Noise) | | | | |
| Fisher Merging | 61.5 | 50.0 | 74.7 | 52.6 | 59.7 | 61.6 | 48.1 | 76.0 | 51.3 | 59.3 |
| RegMean | 66.9 | 51.0 | 80.6 | 68.7 | 66.8 | 69.4 | 41.8 | 84.0 | 67.7 | 65.7 |
| Task Arithmetic | 59.8 | 53.3 | 72.3 | 45.0 | 57.6 | 61.5 | 52.5 | 75.0 | 50.1 | 59.8 |
| Ties-Merging | 60.2 | 45.6 | 69.8 | 38.3 | 53.5 | 61.8 | 47.3 | 73.1 | 42.3 | 56.1 |
| AdaMerging | 69.2 | 40.0 | 79.6 | 83.3 | 68.0 | 70.0 | 53.3 | 82.1 | 80.0 | 71.4 |
| WEMoE | 75.1 | 9.7 | 91.5 | 91.8 | 67.0 | 76.5 | 9.6 | 92.7 | 88.7 | 66.8 |
| | Corrupted Test Set (Pixelate) | | | | | Corrupted Test Set (Spatter) | | | | |
| Fisher Merging | 2.2 | 34.0 | 17.0 | 63.2 | 29.1 | 61.4 | 64.2 | 74.6 | 47.3 | 61.9 |
| RegMean | 2.3 | 38.3 | 18.2 | 89.4 | 37.0 | 67.7 | 60.0 | 81.3 | 81.9 | 72.7 |
| Task Arithmetic | 2.3 | 33.2 | 19.1 | 65.6 | 30.0 | 61.0 | 62.5 | 72.8 | 57.0 | 63.3 |
| Ties-Merging | 3.3 | 31.8 | 18.0 | 58.5 | 27.9 | 61.3 | 52.9 | 70.3 | 48.1 | 58.2 |
| AdaMerging | 1.3 | 52.9 | 21.0 | 91.0 | 41.5 | 68.4 | 55.9 | 78.3 | 92.3 | 73.7 |
| WEMoE | 0.5 | 11.6 | 2.3 | 97.5 | 28.0 | 75.1 | 9.7 | 91.4 | 96.3 | 68.1 |
| | Corrupted Test Set (Contrast) | | | | | Corrupted Test Set (JPEG Compression) | | | | |
| Fisher Merging | 63.8 | 58.4 | 75.5 | 70.4 | 67.0 | 66.3 | 67.6 | 82.6 | 58.9 | 68.8 |
| RegMean | 69.6 | 64.8 | 84.4 | 90.0 | 77.2 | 71.5 | 72.6 | 88.7 | 82.2 | 78.7 |
| Task Arithmetic | 62.3 | 55.7 | 75.3 | 70.8 | 66.0 | 63.9 | 66.1 | 80.1 | 61.0 | 67.8 |
| Ties-Merging | 64.2 | 52.4 | 74.8 | 63.5 | 63.7 | 65.0 | 59.5 | 77.9 | 53.2 | 63.9 |
| AdaMerging | 73.1 | 67.4 | 83.0 | 96.2 | 79.9 | 72.9 | 70.7 | 86.3 | 90.6 | 80.1 |
| WEMoE | 77.2 | 34.7 | 93.1 | 98.4 | 75.9 | 77.3 | 61.0 | 94.1 | 95.7 | 82.0 |

that were not encountered during the training and model fusion process. Additional details and discussions regarding the generalization experiments can be found in Appendix D. (2) Furthermore, in Table 6, we investigate the robustness of multi-task model fusion algorithms by evaluating their performance on corrupted test sets. These corrupted test sets are designed to simulate real-world scenarios where the input data may be noisy or corrupted.

We have the following key observations: (1) The performance of all multi-task model fusion methods on unseen tasks is generally lower than their performance on seen tasks. This is expected, as the models being fused are not explicitly trained on the unseen tasks. (2) A *negative transfer* is observed in Table 17 on the RESISC45 dataset, where the merged models exhibit lower accuracy compared to the pre-trained model. The performance of all multi-task model fusion methods on RESISC45 is lower than the pre-trained model, indicating that the knowledge transferred from the seen tasks may not be beneficial or even harmful to this specific unseen task. (3) The performance of all methods drops significantly on certain types of corruptions, such as pixelation and impulse noise. This highlights the challenge of maintaining robustness under severe distribution shifts and the need for further research in this direction. (4) When the test distribution is corrupted, adaptive methods may overfit to certain tasks, leading to a decrease in overall performance. This suggests that adaptive methods may need to be further regularized to improve generalization and robustness.

## 4.4 APPLYING MODEL FUSION METHODS TO LARGE-SCALE NEURAL NETWORKS

Model fusion methods can also be applied to large-scale neural networks including Large Language Models (LLMs) and Multimodal Large Language Models (MLLMs). The high computational cost associated with developing LLMs are a significant practical challenge for many researchers. In FusionBench, we have developed multiple model fusion techniques applicable to LLMs for cheap and

Table 7: Comparison of individual Mistral-7B models and the upscaled model on various benchmark tasks. For our method, we set $k_{gate} = 8, k = 512$, and the total parameter count is 11.2B. For a better comparison, we also include the performance of the `Qwen1.5-14B` model as a reference.

| MODEL | MMLU | TruthfulQA | GSM8K | ARC Challenge |
|---|---|---|---|---|
| Mistral-7B-v0.1 (pre-trained) | 59.64 | 42.62 | 38.81 | 53.92 |
| Qwen1.5-14B (reference) | 66.11 | 52.00 | 69.37 | 49.93 |
| MetaMath-Mistral-7B | 60.56 | 44.79 | **71.49** | 51.02 |
| dolphin-2.1-mistral-7b | 60.56 | **55.88** | 56.93 | 57.00 |
| speechless-code-mistral-7b-v1.0 | 61.18 | 47.47 | 48.98 | 57.68 |
| Simple Average | **61.42** | 49.95 | 67.40 | 57.59 |
| Task Arithmetic ($\lambda = 0.3$) | 61.29 | 49.38 | 66.94 | **57.94** |
| SMILE Upscaled model ($k = 512$, 11.2B) | 60.66 | 52.79 | 67.85 | 54.35 |
| SMILE Upscaled model (Dense experts) | 60.61 | 54.23 | 70.66 | 55.12 |

efficient model scaling, which can save computational resources for developing new models. We plan to expand the benchmark to include LLM task evaluation in the future. However, currently, this is not our main focus due to the following challenges: (1) **Evaluation immaturity**: The evaluation methodologies for LLMs and MLLMs are not as well-established or standardized as those for the tasks already included in our benchmark. (2) **Resource constraints**: The high computational costs associated with reproducing experiments involving LLMs and MLLMs pose a significant practical challenge for many researchers. Therefore, after merging LLMs using algorithms implemented in FusionBench, we should utilize established evaluation frameworks like `LM-Evaluation-Harness` (Gao et al., 2024) to assess the performance of the fused models on various LLM tasks.

Take merging Mistral-7B models using SMILE upscaling (Tang et al., 2024b) as an example, we compare the performance of the individual Mistral-7B models and the upscaled model on various benchmark tasks in Table 7, as well as the performance of the `Qwen1.5-14B` model as a reference. The `LM-Evaluation-Harness` is utilized to assess the performance of the models. We fuse three Mistral-7B models, each fine-tuned for a distinct downstream task as showing varying performance across different tasks, thereby incorporating task-specific expertise. It is observed that the upscaled models and merged methods (Simple Average, Task Arithmetic) generally enhance performance compared to individual models, demonstrating the benefits of model fusion techniques.

## 5 CONCLUSIONS, FUTURE PLANS

**Conclusions.** We've developed a flexible and modular codebase, which serves as the foundation for *FusionBench*. Our benchmark provides a comprehensive evaluation framework for assessing the performance of multi-task model fusion algorithms. This innovative and comprehensive framework underscores the advantages of a scalable and extendable architecture, thereby simplifying the creation of deep model fusion algorithms. We also organize and provide a collection of datasets and models, which can be utilized to ensure a fair comparison. Last, FusionBench comes with extensive documentation and a series of tutorials, making it user-friendly for beginners and interested researchers. We hope that the community will leverage this benchmark to develop and evaluate new fusion algorithms and to further the popularity of deep model fusion in the machine learning community.

**Limitations and Future Plans.** To date, *FusionBench* primarily focuses on the evaluation of deep model fusion algorithms for multi-task learning. Despite having implemented numerous fusion algorithms, including those that don't primarily focus on multi-task learning, we have not yet to investigate the evaluation for these methods. In the future, we plan to extend the benchmark to provide a more comprehensive evaluation framework for them. What's more, we plan to extend the benchmark by incorporating additional datasets and applications, such as human preference alignment, multi-modal fusion, and reinforcement learning tasks.

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

# A  FLOWCHART OF FUSIONBENCH

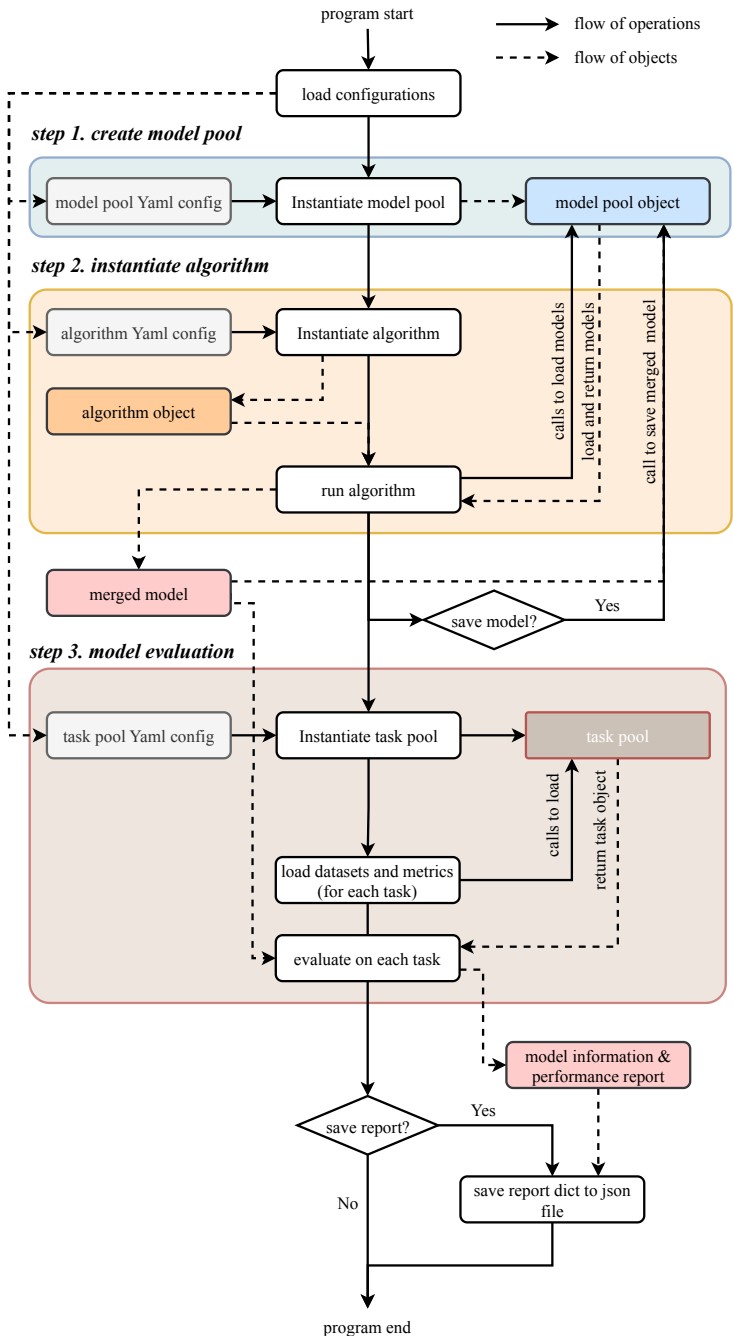

Figure 5: Flowchart of FusionBench.

Figure 5 illustrates the steps to fuse a model using FusionBench across three main stages: creating a model pool, instantiating and runing an algorithm, and (optional) model evaluation.

## B    EVALUATION OF FINE-TUNED SINGLE-TASK MODELS

In this section, we describe the experimental setup for fine-tuning the models and present the performance results of the fine-tuned single-task models.

1. **CLIP-ViT-B/32 Models**: The CLIP models are fine-tuned on eight image classification tasks: SUN397, Cars, RESISC45, EuroSAT, SVHN, GTSRB, MNIST, and DTD. The Adam Optimizer is employed with a fixed learning rate of 1e-5 for a total of 4000 training steps with the batch size of 32. Only the vision encoder is fine-tuned to maintain the model's open-vocabulary characteristic. The performance of fine-tuned CLIP-ViT-B/32 and CLIP-ViT-L/14 models on the eight image classification tasks is shown in Tables 8 and 9, respectively. In Figure 6, we visualize the cosine similarity matrices of task vectors for CLIP-ViT-B/32 and CLIP-ViT-L/14 models. We note that the task vectors for models from various tasks are nearly orthogonal. This suggests that the knowledge specific to each task resides in distinct directions or subspaces. This finding motivates the exploration of locating subspaces in which the knowledge of different tasks can be merged effectively, as discussed in Tang et al. (2023b).

2. **ResNet-50 Models**: We fine-tune ResNet-50 models on three scene understanding tasks: segmentation, depth estimation, and normal estimation using the NYUv2 dataset, each with a learning rate of 1e-4 for 40 epochs, the learning rate is reduced by a factor of 0.5 every 10 epochs. The performance of fine-tuned single-task ResNet-50 models on the NYUv2 dataset is shown in Table 4.

3. **GPT-2 Models**: GPT2 model fine-tuned on tasks from GLUE benchmark, using a constant learning rate of 5e-5 for 3 epochs. The performance of fine-tuned single-task GPT-2 models on the seven text classification tasks is shown in Table 10.

4. **Flan-T5 Models**: In this work, we fine-tune Flan-T5-base and Flan-T5-large models on eight text-to-text generation tasks from the GLUE benchmark. The results of LoRA fine-tuned Flan-T5-base and Flan-T5-large models are shown in Tables 11 and 12, respectively.

Table 8: Performance of fine-tuned single-task CLIP-ViT-B/32 models on the eight image classification tasks.

| MODEL | SUN397 | Cars | RESISC45 | EuroSAT | SVHN | GTSRB | MNIST | DTD |
|---|---|---|---|---|---|---|---|---|
| Pre-trained | 63.2 | 59.8 | 60.7 | 46.0 | 31.6 | 32.5 | 48.2 | 43.9 |
| SUN397 | **75.0** | 47.0 | 54.3 | 46.5 | 28.3 | 26.4 | 44.3 | 41.6 |
| Cars | 56.6 | **78.3** | 50.9 | 38.4 | 30.2 | 30.6 | 49.7 | 41.8 |
| RESISC45 | 52.0 | 47.2 | **95.2** | 56.9 | 23.9 | 24.3 | 39.7 | 35.9 |
| EuroSAT | 49.0 | 39.9 | 33.5 | **99.0** | 11.8 | 22.9 | 33.8 | 35.5 |
| SVHN | 40.5 | 36.3 | 18.9 | 9.8 | **97.3** | 27.3 | 81.8 | 23.2 |
| GRSRB | 36.9 | 33.0 | 20.6 | 21.3 | 41.2 | **98.9** | 30.9 | 23.9 |
| MNIST | 50.3 | 40.0 | 31.3 | 17.7 | 50.1 | 19.3 | **99.6** | 30.7 |
| DTD | 54.6 | 51.3 | 36.8 | 25.0 | 28.9 | 21.8 | 47.3 | **79.7** |

Based on the performance metrics detailed in these tables, we observe that the fine-tuned models demonstrate high accuracy on specific tasks. This observation holds true across various model architectures and task domains, indicating the effectiveness of the fine-tuning process in adapting pre-trained models to excel in particular applications.

What's more, fine-tuning a model on one task can lead to both positive and negative transfer effects on other tasks. Positive transfer occurs when the knowledge gained from fine-tuning on one task enhances the model's performance on another task, while negative transfer arises when the fine-tuning process on one task hinders the model's ability to perform well on other tasks.

Table 9: Performance of fine-tuned single-task CLIP-ViT-L/14 models on the eight image classification tasks.

| MODEL | SUN397 | Cars | RESISC45 | EuroSAT | SVHN | GTSRB | MNIST | DTD |
|---|---|---|---|---|---|---|---|---|
| Pre-trained | 68.3 | 77.8 | 71.0 | 58.9 | 58.4 | 50.6 | 76.4 | 55.5 |
| SUN397 | **82.8** | 68.4 | 58.1 | 49.9 | 55.0 | 46.3 | 79.5 | 52.8 |
| Cars | 67.8 | **92.9** | 68.7 | 56.4 | 51.7 | 47.7 | 80.5 | 55.6 |
| RESISC45 | 65.6 | 69.0 | **97.4** | 64.3 | 38.3 | 46.6 | 77.7 | 49.9 |
| EuroSAT | 65.2 | 69.0 | 40.6 | **99.2** | 33.4 | 45.6 | 73.5 | 47.1 |
| SVHN | 66.5 | 69.0 | 54.0 | 19.7 | **97.9** | 48.7 | 92.2 | 50.1 |
| GRSRB | 63.4 | 64.8 | 38.7 | 19.6 | 71.0 | **99.2** | 75.1 | 45.8 |
| MNIST | 56.1 | 49.8 | 53.5 | 26.6 | 48.2 | 33.1 | **99.8** | 47.1 |
| DTD | 66.8 | 75.3 | 65.5 | 43.7 | 49.5 | 45.0 | 68.5 | **85.5** |

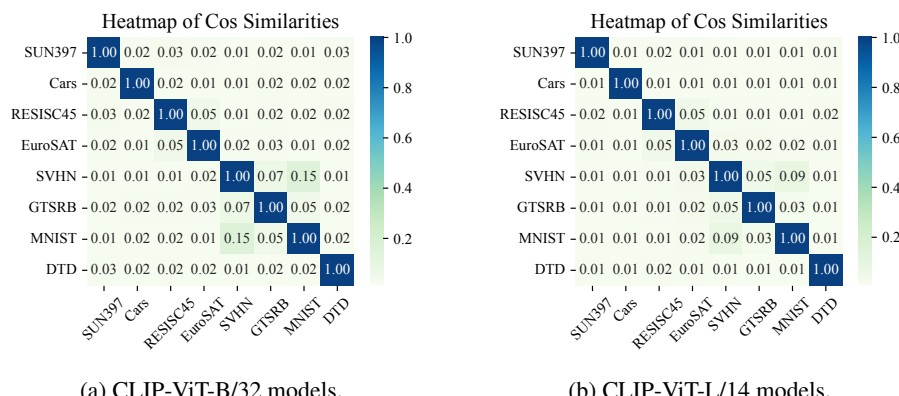

(a) CLIP-ViT-B/32 models.  (b) CLIP-ViT-L/14 models.

Figure 6: Cosine similarity matrices of task vectors for CLIP-ViT-B/32 and CLIP-ViT-L/14 models.

Table 10: Performance of fine-tuned single-task GPT-2 models on the seven text classification tasks.

| MODEL | CoLA | MNLI | MRPC | QNLI | QQP | RTE | SST-2 |
|---|---|---|---|---|---|---|---|
| CoLA | **76.8** | 32.8 | 68.4 | 50.4 | 39.2 | 48.0 | 51.0 |
| MNLI | 59.5 | **82.1** | 33.8 | 46.5 | 24.9 | 57.4 | 40.5 |
| MRPC | 30.8 | 25.9 | **80.4** | 47.1 | 65.9 | 49.1 | 49.1 |
| QNLI | 58.7 | 38.9 | 30.6 | **88.3** | 39.9 | 48.7 | 47.0 |
| QQP | 31.4 | 25.7 | 62.3 | 45.0 | **89.6** | 49.1 | 49.1 |
| RTE | 52.8 | 47.7 | 37.5 | 53.5 | 33.7 | **65.3** | 54.9 |
| SST-2 | 51.8 | 32.9 | 40.2 | 49.8 | 56.8 | 44.4 | **91.2** |

Table 11: Performance of LoRA fine-tuned Flan-T5-Base models on the eight text-to-text generation tasks from the GLUE benchmark.

| MODEL | CoLA | MNLI | MRPC | QNLI | QQP | RTE | SST2 | STSB |
|---|---|---|---|---|---|---|---|---|
| Pre-trained | 69.1 | 56.5 | 76.2 | 88.4 | 82.1 | 80.1 | 91.2 | 62.2 |
| CoLA | 69.1 | 39.9 | 75.2 | 89.1 | 81.1 | 81.9 | 90.7 | 54.0 |
| MNLI | **69.4** | **82.7** | 73.8 | 89.3 | 82.0 | 79.4 | 90.9 | 68.1 |
| MRPC | 64.0 | 44.9 | **85.5** | 82.6 | 81.0 | 69.0 | 88.6 | 73.6 |
| QNLI | 68.9 | 52.7 | 76.7 | **90.9** | 82.8 | 79.8 | 91.5 | 68.9 |
| QQP | 65.0 | 54.6 | 75.7 | 89.0 | **84.0** | 81.6 | 90.7 | 75.3 |
| RTE | 64.9 | 51.8 | 69.4 | 89.2 | 79.8 | **84.5** | 90.6 | 70.1 |
| SST2 | 68.3 | 56.6 | 76.0 | 88.5 | 83.4 | 79.8 | **92.9** | 62.6 |
| STSB | 65.7 | 1.7 | 67.4 | 89.3 | 80.1 | 79.8 | 90.8 | **87.4** |

Table 12: Performance of LoRA fine-tuned Flan-T5-Large models on the eight text-to-text generation tasks from the GLUE benchmark.

| MODEL | CoLA | MNLI | MRPC | QNLI | QQP | RTE | SST2 | STSB |
|---|---|---|---|---|---|---|---|---|
| Pre-trained | 73.7 | 56.6 | 82.4 | 91.1 | 85.5 | 85.6 | 94.3 | 87.5 |
| CoLA | **80.2** | 53.9 | 81.4 | 90.8 | 84.5 | 84.1 | 93.9 | 87.1 |
| MNLI | 73.7 | **88.5** | 77.9 | 92.4 | 85.2 | 87.7 | 94.4 | 86.7 |
| MRPC | 75.6 | 52.6 | **89.2** | 92.6 | 84.4 | 86.3 | 94.3 | 86.3 |
| QNLI | 73.5 | 54.5 | 82.8 | **94.4** | 85.8 | 85.2 | 93.7 | 87.1 |
| QQP | 74.0 | 53.8 | 82.8 | 92.5 | **87.2** | 85.6 | 94.5 | 88.3 |
| RTE | 75.6 | 57.5 | 69.9 | 92.8 | 83.8 | **91.7** | 94.6 | 86.0 |
| SST2 | 73.6 | 55.3 | 82.1 | 91.6 | 85.5 | 85.2 | **95.2** | 86.9 |
| STSB | 73.4 | 39.3 | 82.1 | 92.6 | 86.1 | 83.4 | 94.0 | **90.9** |

## C   MULTI-TASK MODEL FUSION

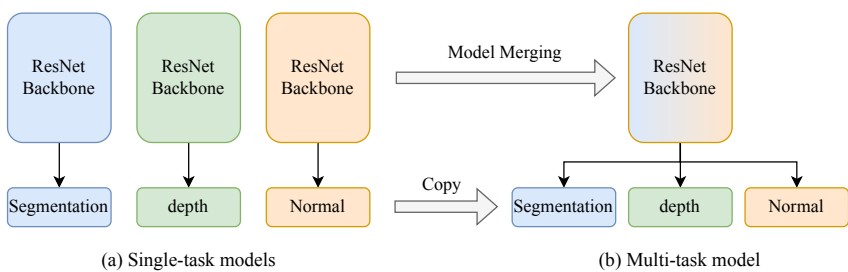

(a) Single-task models    (b) Multi-task model

Figure 7: Merging ResNet-50 models on three scene understanding tasks: segmentation, depth estimation, and normal estimation. Where the backbones are merged and the heads are kept separate.

Table 13: Multi-task performance when merging GPT-2 models on seven text classification tasks.

| METHOD | CoLA | MNLI | MRPC | QNLI | QQP | RTE | SST-2 | Avg. |
|---|---|---|---|---|---|---|---|---|
| *Reference Methods* | | | | | | | | |
| Fine-tuned (STL) | 76.8 | 82.1 | 80.4 | 88.3 | 89.6 | 65.3 | 91.2 | 82.0 |
| *Multi-Task Model Fusion Method* | | | | | | | | |
| Simple Average | 55.0 | 55.1 | 51.0 | 57.6 | 76.7 | 44.8 | 52.5 | 56.1 |
| Fisher Merging | 54.8 | 58.0 | 39.5 | 63.3 | 81.5 | 49.1 | 64.7 | 58.7 |
| RegMean | 61.7 | 70.4 | 65.4 | 69.7 | 78.8 | 56.0 | 79.7 | 68.8 |
| Task Arithmetic | 68.7 | 68.6 | 69.6 | 70.5 | 81.8 | 47.3 | 83.6 | 70.0 |
| Ties-Merging | 68.4 | 71.4 | 68.4 | 69.6 | 82.4 | 47.7 | 81.8 | 70.0 |

We begin by comparing the performance of multi-task model fusion algorithms on various tasks using different models. These experiments provide insights into the effectiveness of different fusion methods in improving the performance of multi-task models. We evaluate the performance of multi-task model fusion algorithms on image classification tasks using CLIP models, scene understanding tasks using ResNet-50 models, text classification tasks using GPT-2 models, and text-to-text generation tasks using Flan-T5 models.

1. **Image Classification Tasks with CLIP Models**: We utilize the CLIP-ViT-B/32 and CLIP-ViT-L/14 models from the HuggingFace Library (Ilharco et al., 2021). The results of merging CLIP-ViT-B/32 models on all eight tasks are provided in Table 3 and Figure 4. The results of CLIP-ViT-L/14 models are shown in Table 15.
2. **Scene Understanding Tasks with ResNet-50 Models**: We use the NYUv2 dataset and ResNet-50 models for segmentation, depth estimation, and normal estimation tasks. In

Table 14: Experimental results of merging Flan-T5-base (LoRA fine-tuned) models on all eight tasks.

| METHOD | CoLA | MNLI | MRPC | QNLI | QQP | RTE | SST2 | STSB | Avg. |
|---|---|---|---|---|---|---|---|---|---|
| *Reference Methods* | | | | | | | | | |
| Pre-trained | 69.1 | 56.5 | 76.2 | 88.4 | 82.1 | 80.1 | 91.2 | 62.2 | 75.7 |
| Individual | 69.1 | 82.7 | 85.5 | 90.9 | 84.0 | 84.4 | 92.9 | 87.4 | 84.6 |
| *Multi-Task Model Fusion Methods* | | | | | | | | | |
| Weight Averaging | 69.7 | 59.7 | 78.9 | 90.1 | 83.8 | 80.5 | 91.2 | 72.0 | 78.2 |
| Task Arithmetic | 68.8 | 55.2 | 78.7 | 89.8 | 83.7 | 79.1 | 91.5 | 72.4 | 77.4 |
| Ties-Merging | 68.3 | 56.3 | 79.4 | 89.8 | 83.7 | 79.4 | 91.6 | 71.2 | 77.5 |
| Layer-wise AdaMerging | 69.1 | 60.3 | 78.4 | 90.0 | 83.6 | 79.1 | 91.6 | 74.1 | 78.3 |
| SMILE (Model Mixing) | 69.3 | 82.9 | 83.8 | 90.6 | 83.9 | 83.4 | 93.1 | 85.1 | 84.0 |

Table 15: Multi-task performance when merging CLIP-ViT-L/14 models on all eight tasks.

| METHOD | SUN397 | Cars | RESISC45 | EuroSAT | SVHN | GTSRB | MNIST | DTD | Avg. |
|---|---|---|---|---|---|---|---|---|---|
| *Reference Methods* | | | | | | | | | |
| Pre-trained | 68.3 | 77.8 | 71.0 | 58.9 | 58.4 | 50.6 | 76.4 | 55.5 | 64.6 |
| Individual Fine-tuned | 82.8 | 92.9 | 97.4 | 99.2 | 97.9 | 99.2 | 99.8 | 85.5 | 94.3 |
| Traditional MTL | 79.0 | 89.3 | 94.5 | 98.4 | 96.4 | 98.1 | 99.4 | 83.7 | 92.4 |
| *Multi-Task Model Fusion Methods* | | | | | | | | | |
| Weight Averaging | 72.5 | 81.5 | 82.2 | 90.0 | 81.6 | 74.0 | 96.6 | 61.8 | 80.0 |
| Fisher Merging | 70.6 | 79.4 | 84.1 | 98.1 | 74.7 | 85.0 | 89.5 | 61.0 | 80.3 |
| RegMean | 75.3 | 88.4 | 90.0 | 97.1 | 95.9 | 92.4 | 98.5 | 72.6 | 88.8 |
| Task Arithmetic | 72.0 | 79.0 | 80.5 | 86.0 | 87.5 | 83.5 | 98.0 | 58.8 | 80.7 |
| Ties-Merging | 74.7 | 83.3 | 86.4 | 91.3 | 89.7 | 85.2 | 97.8 | 63.9 | 84.0 |
| task-wise AdaMerging | 75.8 | 80.1 | 77.2 | 83.6 | 68.4 | 93.5 | 93.1 | 69.0 | 80.1 |
| layer-wise AdaMerging | 78.1 | 90.7 | 90.8 | 96.5 | 94.8 | 97.5 | 98.6 | 81.3 | 91.0 |
| WEMoE (Model Mixing) | 81.5 | 92.3 | 96.5 | 98.8 | 97.6 | 99.4 | 99.6 | 84.5 | 93.8 |
| SMILE (Model Mixing) | 81.9 | 92.3 | 95.5 | 99.1 | 98.0 | 98.9 | 99.7 | 83.6 | 93.6 |

Figure 7, we illustrate the process of merging ResNet-50 models on these tasks, where the backbones are merged, and the heads are copied separately. The results of merging ResNet-50 models on these tasks are shown in Table 4.

3. **Text Classification Tasks with GPT-2 Models**: The results of merging GPT-2 models on seven text classification tasks are shown in Table 13.

4. **Text-to-Text Generation Tasks with Flan-T5 Models**: For LoRA fine-tuned Flan-T5-base and Flan-T5-large models, we merge and unload the LoRA adapters before performing multi-task model fusion. The results of merging Flan-T5-base and Flan-T5-large models on all eight tasks are shown in Tables 14 and 16, respectively.

In the above mentioned tables, we compare the performance of different multi-task model fusion algorithms on various tasks. The results of pre-trained models, fine-tuned models, and traditional multi-task learning (MTL) are provided as reference methods.

Table 16: Experimental results of merging Flan-T5-large (LoRA fine-tuned) models on all eight tasks.

| METHOD | CoLA | MNLI | MRPC | QNLI | QQP | RTE | SST2 | STSB | Avg. |
|---|---|---|---|---|---|---|---|---|---|
| *Reference Methods* | | | | | | | | | |
| Pre-trained | 73.7 | 56.6 | 82.4 | 91.1 | 85.5 | 85.6 | 94.3 | 87.5 | 82.1 |
| Individual | 80.2 | 88.5 | 89.2 | 94.4 | 87.2 | 91.7 | 95.2 | 90.9 | 89.6 |
| *Multi-Task Model Fusion Methods* | | | | | | | | | |
| Weight Averaging | 74.6 | 84.3 | 84.1 | 92.8 | 86.3 | 87.4 | 94.8 | 88.0 | 86.5 |
| Task Arithmetic | 76.9 | 85.4 | 85.3 | 93.9 | 85.8 | 88.1 | 95.2 | 87.8 | 87.3 |
| Ties-Merging | 77.1 | 85.1 | 86.3 | 93.9 | 86.0 | 87.7 | 95.1 | 88.0 | 87.4 |
| Layer-wise AdaMerging | 76.7 | 87.6 | 84.8 | 93.8 | 85.9 | 88.1 | 95.2 | 88.6 | 87.6 |

# D GENERALIZATION EXPERIMENTS

Table 17: Generalization results on two unseen tasks when merging ViT-B/32 models on six tasks.

| METHOD | Seen Tasks (ACC) | | | | | | | Unseen Tasks (ACC) | | |
|---|---|---|---|---|---|---|---|---|---|---|
| | SUN397 | Cars | GTSRB | EuroSAT | DTD | MNIST | Avg. | RESISC45 | SVHN | Avg. |
| Pre-trained | 63.2 | 59.9 | 30.4 | 45.6 | 43.9 | 47.6 | 48.4 | 60.6 | 23.5 | 40.1 |
| Fisher Merging | 68.1 | 67.4 | 67.2 | 86.4 | 58.6 | 81.6 | 71.5 | 60.2 | 42.5 | 51.3 |
| RegMean | 69.4 | 70.5 | 86.9 | 97.0 | 67.1 | 98.3 | 81.5 | 50.2 | 51.5 | 50.8 |
| Task Arithmetic | 65.2 | 63.6 | 76.1 | 87.1 | 56.4 | 94.2 | 73.8 | 52.4 | 45.2 | 48.8 |
| Ties-Merging | 68.2 | 65.9 | 70.0 | 81.2 | 56.0 | 89.0 | 71.7 | 60.3 | 47.3 | 53.8 |
| AdaMerging | 69.8 | 72.4 | 95.5 | 95.1 | 70.7 | 98.1 | 83.6 | 48.7 | 60.7 | 54.7 |
| WEMoE | 74.3 | 78.1 | 98.8 | 98.7 | 75.1 | 99.5 | 87.4 | 47.3 | 51.3 | 49.3 |

For the generalization experiments, we assess the performance of multi-task model fusion algorithms on two unseen tasks after merging ViT-B/32 models trained on six tasks. The performance of various multi-task model fusion methods, including Fisher Merging (Matena & Raffel, 2022), RegMean (Jin et al., 2022), Task Arithmetic (Ilharco et al., 2022), Ties-Merging (Yadav et al., 2023), AdaMerging (Yang et al., 2023), and WEMoE (Tang et al., 2024c), is compared across both the seen tasks and unseen tasks.

Specifically, we conduct two sets of generalization experiments using the CLIP-ViT-B/32 models:

- In the first set, we merge models trained on six tasks (SUN397, Cars, RESISC45, DTD, SVHN, GTSRB) and evaluate the fused model on the unseen tasks (MNIST, EuroSAT). The results are shown in Table 5.

- In the second set of experiments, we merge models trained six tasks (SUN397, Cars, GTSRB, EuroSAT, DTD, MNIST) and evaluate the fused model on the unseen tasks (RESISC45, SVHN). The results are shown in Table 17.

By conducting these two sets of generalization experiments, we aim to gain a comprehensive understanding of how the CLIP-ViT-B/32 models, when fused with knowledge from different task combinations, can perform on various unseen tasks. From these experimental results, we can observe instances of negative transfer when evaluating the fused CLIP models on unseen tasks. Here, negative transfer occurs when the knowledge gained from fine-tuning on a set of tasks hinders the model's performance on new, unseen tasks. In other words, the model's ability to generalize and adapt to novel challenges is compromised due to the specific knowledge acquired during the fine-tuning process. The presence of negative transfer in these experiments highlights the challenges and limitations of model fusion and generalization in the context of the CLIP-ViT-B/32 models. Several factors can contribute to negative transfer, such as:

- *Task dissimilarity*: If the unseen tasks are significantly different from the tasks used for fine-tuning, the learned representations may not be directly applicable, leading to performance degradation.

- *Overspecialization*: Fine-tuning on a specific set of tasks may cause the model to overfit to task-specific features and patterns, reducing its ability to generalize to new tasks.

- *Interference between tasks*: When merging knowledge from multiple tasks, there may be conflicts or interference between the learned representations, hindering the model's ability to adapt to unseen tasks effectively.

To mitigate the negative transfer and improve the generalization ability of merged models, several strategies can be explored, such as:

- *Task selection*: Carefully selecting tasks that are more similar or complementary to the target unseen tasks can help reduce the risk of negative transfer. This is adapted in Wu et al. (2023), where the Fisher information matrix is computed for a proxy metric for task similarity.

- *Regularization techniques*: Applying regularization methods, such as weight decay or dropout, during the fine-tuning process may help prevent overfitting and promote better generalization.

## E  PROMPT-BASED TEXT-TO-TEXT GENERATION

This section details the prompt templates employed for each of the eight text-to-text generation tasks from the GLUE benchmark., see Section 3.3 for more details. Within each task, we provide the format of the input text, and the corresponding target text mapping. These templates are crucial in fine-tuning the Flan-T5 models for generating appropriate text outputs tailored to each specific task.

- CoLA:
  - *Input Text:* "Indicate if the following sentence is grammatically correct or not: "sentence". Answer 'acceptable' or 'unacceptable'."
  - *Target Text:*
    * 0: "unacceptable"
    * 1: "acceptable"
- MNLI:
  - Input Text: "Does the premise: 'premise' logically imply, contradict, or is neutral to the hypothesis: 'hypothesis'? Answer with 'entailment', 'contradiction', or 'neutral'."
  - Target Text:
    * 0: "entailment"
    * 1: "neutral"
    * 2: "contradiction"
- MRPC:
  - *Input Text:* "Are the following sentences 'sentence1' and 'sentence2' conveying the same meaning? Answer with 'yes' or 'no'."
  - *Target Text:*
    * 0: "no"
    * 1: "yes"
- QNLI:
  - *Input Text:* "Given the context: 'sentence', does the question 'question' have an answer based on the information provided? Answer with 'yes' or 'no'."
  - *Target Text:*
    * 0: "yes"
    * 1: "no"
- QQP:

- *Input Text:* "Do the questions 'question1' and 'question2' have the same intent? Answer with 'yes' or 'no'."
- *Target Text:*
  * 0: "no"
  * 1: "yes"

- RTE:
  - *Input Text:* "Does the text: 'sentence1' entail that 'sentence2' is true? Provide 'yes' or 'no'."
  - *Target Text:*
    * 0: "yes"
    * 1: "no"

- SST-2:
  - *Input Text:* "Given the sentence 'sentence', determine the sentiment. Is it positive or negative?"
  - *Target Text:*
    * 0: "negative"
    * 1: "positive"

- STSB:
  - *Input Text:* "Consider the sentences 'sentence1' and 'sentence2'. On a scale from 1 (completely different) to 5 (completely similar), rate the similarity."
  - *Target Text:* ":.1f", parse to float with one decimal place

**Reporting Metrics:** We report accuracy for all tasks except for STSB, where we use Spearman's $\rho$ as the evaluation metric. For task STSB, the model is expected to output a numerical value. An example from the STSB task is as follows:

- *Input*:
  - Sentence 1: A plane is taking off.
  - Sentence 2: An air plane is taking off.
- *Output*:
  - label: 5

We try to parse the output as a numerical value. If the model outputs a numerical value, we can calculate the Spearman's rho between the predicted numerical value and the ground truth numerical value. If the model outputs a non-numerical value, we assume the Spearman's rho is 0, indicating that there is no discernible monotonic increasing or decreasing relationship between the model's predictions and the ground truth. This is a conservative approach, as even non-numerical outputs might contain some relevant information that's being discarded in this evaluation.

