# OpenReview forum: "FusionBench: A Comprehensive Benchmark of Deep Model Fusion"
_ICLR.cc/2025/Conference — Submitted to ICLR 2025_

### Official Review · Reviewer_W4dN · 2024-11-01

**Soundness:** 2
**Presentation:** 3
**Contribution:** 2
**Rating:** 3
**Confidence:** 4

**Summary:**

The paper presents FusionBench, a benchmark for evaluating deep model fusion techniques, which unify multiple neural networks into a single, more effective model.

**Strengths:**

1- The proposed pipeline offers various tasks (image and text classification, text generation) and includes multiple models and fusion strategies (e.g., model ensembles, merging, and mixing).

2- FusionBench standardises evaluations across tasks, model sizes, and fine-tuning methods, providing resources like documentation and tutorials to support researchers in replicating results.

3- The paper is well-written, making it easy to read and understand.

**Weaknesses:**

1- This work appears to be an engineering effort to generalise existing models and methods within a unified pipeline. However, it lacks scientific novelty and significant technological advancement. Therefore, it may be better suited for applied venues, such as workshops, rather than a top-tier fundamental research conference.

2- Besides the above issue, it still seems unclear how the model mixing happens in the background, though they state a list of existing methods they use in Table 1.

3- Figure 1 is not very illustrative as Fig-1(b) and Fig-1(c) are almost identical. Further demonstration might be required in this regard.

4- In terms of evaluation, it would be more beneficial to see how the proposed system works on large-scale data, such as the ImageNet-1k involved.

5- I could not find any evidence of the claimed deliverable in the submitted work as stated in the abstract. "...This includes detailed documentation, code examples, and tutorials, making FusionBench a user-friendly and accessible platform for both beginners and experienced researchers."

**Questions:**

Please refer to the Weaknesses section.

---

> ### Author Response · Authors · 2024-11-14
>
> **W1: This work appears to be an engineering effort to generalise existing models and methods within a unified pipeline. However, it lacks scientific novelty and significant technological advancement. Therefore, it may be better suited for applied venues, such as workshops, rather than a top-tier fundamental research conference.**
>
> Thank you for your thoughtful review and for highlighting your concerns.
> While our work does involve engineering efforts to unify existing models and methods, we believe it goes beyond mere integration and introduces significant scientific advancements.
>
> - Our work enables the systematic investigation of model fusion techniques across a wide range of tasks and domains.
> - Our work enables reproducible research through standardized benchmarking. This facilitates comparative studies across different fusion approaches.
> - **Research Impact:** There are already several studies that are based on the codebase of FusionBench, including research on more advanced model fusion techniques and investigations into trustworthy model fusion methods.
>
> **W2: Besides the above issue, it still seems unclear how the model mixing happens in the background, though they state a list of existing methods they use in Table 1.** and
> **W3: Figure 1 is not very illustrative as Fig-1(b) and Fig-1(c) are almost identical. Further demonstration might be required in this regard.**
>
> We appreciate the feedback. The complexity of model mixing techniques presents significant challenges for clear visual representation, primarily due to the architecture variability - model mixing methods fundamentally alter the model architecture in distinct ways. Specific Technical Challenges are as follows:
>
> - Depth Upscaling modifies the number of layers in the network resulting in variable depth structures.
> - Model Expansion methods alter the dimensional properties of hidden states. This changes the internal representation spaces and introduces new parameter matrices of different sizes.
> - Mixture-of-Experts (MoE) Upscaling transforms standard dense layers into complex MoE structures. This introduces routing mechanisms and expert networks. Creates branching architectures that are inherently more complex.
>
> Therefore, creating a unified visual representation would require multiple different diagrams.
> The readers can refer to the online documentation for more detailed explanations of each model mixing technique in this [anonymous link](https://anonymous.4open.science/w/fusion_bench-gh-page-A360/algorithms/depth_upscaling/).
>
> **W4: In terms of evaluation, it would be more beneficial to see how the proposed system works on large-scale data, such as the ImageNet-1k involved.**
>
> We appreciate the suggestion.
> In fact, it is convenient to conduct evaluations on custom image classification datasets using FusionBench.
> We provide additional examples of how to configure and run the benchmark on custom datasets in the codebase.
> [Anonymous links to the example configurations can be found here](https://anonymous.4open.science/r/fusion_bench-79EC/config/dataset/image_classification/train/tiny-imagenet.yaml).
>
> In this benchmark, we are currently focusing on evaluating the performance of multi-task model fusion in the in-domain multi-task setting.
>
> **W5:  I could not find any evidence of the claimed deliverable in the submitted work as stated in the abstract. "...This includes detailed documentation, code examples, and tutorials, making FusionBench a user-friendly and accessible platform for both beginners and experienced researchers."**
>
> Due to the double-blind review process, we can not provide a direct link to the online documentation in the paper.
> Here is the anonymous link to the online documentation for the FusionBench project: https://anonymous.4open.science/w/fusion_bench-gh-page-A360/

---

> > ### Comment · Reviewer_W4dN · 2024-11-27
> > **response to authors feedback**
> >
> > Although the authors' responses partially address my comments regarding the details of the work, they still fall short in demonstrating research novelty. In my view, this work represents a valuable engineering effort that could benefit the community; however, the authors' arguments do not elevate it to the level of fundamental research. Additionally, there are already many tools (e.g., Detectron2, MMDetection) designed to consolidate state-of-the-art algorithms, facilitate their use by the community, and support rapid implementation and evaluation of research. While such efforts are commendable, they do not constitute groundbreaking research. From this perspective, while I acknowledge the authors' efforts in developing this platform, it does not meet the criteria for a research contribution. Therefore, I will maintain my decision to reject.

---

### Official Review · Reviewer_i5gE · 2024-11-04

**Soundness:** 3
**Presentation:** 3
**Contribution:** 2
**Rating:** 6
**Confidence:** 2

**Summary:**

The paper proposed the first deep model fusion benchmark FusionBench together with codebase. The codebase is composed of three main modules: Algorithm, Mode Pool, and Task Pool. FusionBench has 16 built-in fusion algorithms implemented. Their comprehensive evaluation results are presented in the paper.

**Strengths:**

1. The paper is very well motivated. As the first evaluation benchmark for deep model fusion, it would surely benefit the future researches in the domain.
2. Extendability is a very important aspect for open-source benchmark. Glad to see it has been taken into consideration.

**Weaknesses:**

Recently, LLM and T2I/V models have gained tremendous attention in the research community. It might be worth adding evaluation for that. E.g., fusion methods for LLaMA based models, StableDiffusion models.

**Questions:**

N/A

**Details Of Ethics Concerns:**

The submission failed to follow the rule of "Anonymous Url: I certify that there is no URL (e.g., github page) that could be used to find authors’ identity." In the submitted codebase as Supplementary Material, there are author's name in the LICENSE file, and non-anonymous GitHub and arXiv links in "docs/README.md".

---

> ### Author Response · Authors · 2024-11-14
>
> We appreciate the suggestions. In fact, we have implemented some fusion algorithms for large language models (LLM) in FusionBench, such as SLERP, DARE-Task Arithmetic, AdaMerging for Llama, Depth Upscaling, and ExPO, etc.
> Anonymous links to the code are listed as follows:
>
> - [SLERP](https://anonymous.4open.science/r/fusion_bench-79EC/fusion_bench/method/slerp/slerp.py)
> - [DARE-Task Arithmetic](https://anonymous.4open.science/r/fusion_bench-79EC/fusion_bench/method/dare/task_arithmetic.py)
> - [AdaMerging for Llama](https://anonymous.4open.science/r/fusion_bench-79EC/fusion_bench/method/adamerging/llama_adamerging.py)
> - [Depth Upscaling](https://anonymous.4open.science/r/fusion_bench-79EC/fusion_bench/method/depth_upscaling/depth_upscaling.py)
> - [ExPO](https://anonymous.4open.science/r/fusion_bench-79EC/fusion_bench/method/linear/expo.py)
>
> But the implementation of these methods is not yet fully tested and documented. We will continue to improve the documentation and testing of these methods.

---

> > ### Comment · Reviewer_i5gE · 2024-11-15
> >
> > Thanks for sharing that.

---

### Official Review · Reviewer_ACwH · 2024-11-04

**Soundness:** 3
**Presentation:** 2
**Contribution:** 3
**Rating:** 5
**Confidence:** 4

**Summary:**

The authors propose a comprehensive benchmark for deep model fusion that evaluates various models across a wide range of CV and NLP tasks. They also develop a modular codebase for user-friendly realization and evaluation of different model fusion approaches.

**Strengths:**

1. The paper is well-written and easy to follow.

2. The proposed framework is modular and user-friendly.

3. The authors comprehensively evaluated deep model fusion across different tasks and models.

**Weaknesses:**

1. While the authors seem to evaluate against 26 tasks as claimed in the paper, I found the actual number of tasks quite limited. For example, there are 8 different datasets used for image classification, but the authors deem them as 8 tasks. Fairly speaking, they can only be regarded as 8 benchmarks on the image classification task. In that sense, the wordings in such as Table 5 should be seen/unseen domains instead of tasks.

2. For the scene understanding part, the tasks are all limited to the dense per-pixel prediction type (depth, normal, and segmentation), and are only conducted against the NYUv2 dataset. I would recommend adding additional tasks that are common in this area, such as object detection.

**Questions:**

I appreciate the authors' engineering efforts in putting together such a framework for evaluating deep model fusion. If the authors can address my concerns properly, I would be happy to edit the rating.

1. In Table 2, the term `tasks` seems abused. Technically, there are 8 datasets instead of tasks for the image classification task. The same applies to the text classification and generation task, with 7 and 8 datasets, respectively. The scene understanding does contain 3 tasks (segmentation, depth, and normal). I recommend the authors edit the terminologies used and make the tables clearer.
- for the image classification task, the setting looks more like domain transfer and generalization
- for the scene understanding, it is standard multi-task learning

The authors may argue that among the 8 datasets, there are object recognition, satellite image classification, etc. "tasks" (what is said in their paper). However, these are all under the open-vocabulary image classification task, and getting down to that granularity does not make sense if we consider the other tasks in the paper (such as depth, segmentation, etc.)

2. All the scene understanding tasks are dense (per-pixel) prediction tasks. It would be interesting to consider tasks such as object detection. Also, using the NYUv2 dataset only covers the in-domain multi-task setting and is limited. How about doing a cross-domain multi-task setting? For example, a segmentation model on ADE20K, plus a depth model on KITTI.

---

> ### Author Response · Authors · 2024-11-14
>
> **W1: While the authors seem to evaluate against 26 tasks as claimed in the paper, I found the actual number of tasks quite limited. For example, there are 8 different datasets used for image classification, but the authors deem them as 8 tasks. Fairly speaking, they can only be regarded as 8 benchmarks on the image classification task. In that sense, the wordings in such as Table 5 should be seen/unseen domains instead of tasks.**
>
> We appreciate the feedback. We have made the necessary changes in Table 2 to the terminology used in the tables to avoid confusion.
>
> - In the context of image classification, we acknowledge that we have used "task" and "domain" interchangeably, which may cause some confusion.
> - For text generation tasks, we maintain a more precise distinction where "task" specifically refers to fundamentally different operations - such as question answering, text summarization, sentiment classification, etc.
>
> The term "domain" would indeed be more appropriate when discussing variations within a single type of task, such as open-vocabulary image classification and  different applications in mathematics or coding challenges.
>
> **W2: For the scene understanding part, the tasks are all limited to the dense per-pixel prediction type (depth, normal, and segmentation), and are only conducted against the NYUv2 dataset. I would recommend adding additional tasks that are common in this area, such as object detection.**
>
> Thank you for the suggestions. We are consistently working on expanding the FusionBench to include more tasks and datasets.
>
> **Q1: In Table 2, the term `tasks` seems abused. Technically, there are 8 datasets instead of tasks for the image classification task. The same applies to the text classification and generation task, with 7 and 8 datasets, respectively. The scene understanding does contain 3 tasks (segmentation, depth, and normal). I recommend the authors edit the terminologies used and make the tables clearer.**
>
> We appreciate the feedback. We have made the necessary changes in Table 2 to the terminology used in the tables to avoid confusion.
>
> **Q2: All the scene understanding tasks are dense (per-pixel) prediction tasks. It would be interesting to consider tasks such as object detection. Also, using the NYUv2 dataset only covers the in-domain multi-task setting and is limited. How about doing a cross-domain multi-task setting? For example, a segmentation model on ADE20K, plus a depth model on KITTI.**
>
> We appreciate the suggestions.
> In fact, many implemented algorithms are model-agnostic and task-agnostic, and can be applyed to object detection tasks with minimal modifications.
>
> Of course, we will also consider adding object detection tasks and cross-domain multi-task settings in the future.

---

### Official Review · Reviewer_qs86 · 2024-11-05

**Soundness:** 3
**Presentation:** 3
**Contribution:** 3
**Rating:** 6
**Confidence:** 4

**Summary:**

This paper presents FusionBench, a comprehensive benchmark platform to evaluate deep model fusion techniques across tasks in image classification, text classification, and text generation. FusionBench organizes fusion techniques into model ensembles, model merging, and model mixing, and includes 26 tasks, 74 fine-tuned models, and 19 fusion algorithms. The benchmark aims to standardize evaluations, ensuring fair comparisons and supporting both novices and researchers with accessible resources and tutorials.

**Strengths:**

1. FusionBench covers a broad range of tasks and fusion methods, providing flexibility for various fusion scenarios in NLP and vision tasks.
2. Accessible resources, detailed documentation, and examples make FusionBench highly usable for beginners and researchers alike.
3. The benchmark offers a fair, standardized evaluation of fusion methods, supporting clear comparisons across tasks.
4. Well-organized structure, visual aids, and concise descriptions make it easy to understand and navigate the platform.

**Weaknesses:**

1. The absence of direct comparisons with baseline models.
2. A deeper explanation for the categorization of model ensembles, merging, and mixing would help clarify their task-specific benefits.
3. More insights on scalability to additional tasks or modalities would enhance FusionBench’s applicability to diverse research needs.
4. FusionBench’s effectiveness depends on high-quality pre-trained models, which might limit performance in under-resourced domains.

**Questions:**

Q1. Did you explore any non-fusion baseline models to provide a more direct comparison of fusion performance benefits? If so, could those results be included to clarify the relative impact of fusion methods?
Q2. Do you foresee expanding FusionBench to include multi-modal tasks, or would that require significant modifications to the existing benchmark structure?
Q3. Are there recommendations for tuning fusion methods based on specific task types, especially for users who may be new to model fusion?
Q4. How does FusionBench handle variability in pre-trained model quality? Are there mechanisms to assess or adjust for model quality across tasks?

---

> ### Author Response · Authors · 2024-11-14
>
> **W1: The absence of direct comparisons with baseline models.**
>
> We include the performance of task-specific CLIP vision models, GPT-2 models and Flan-T5 models in the supplementary material. It can also be found in our online documentation in the [anonymous links](https://anonymous.4open.science/w/fusion_bench-gh-page-A360/modelpool/clip_vit/#performance-of-the-fine-tuned-models).
>
> For example, for CLIP-ViT-B/32 models, here are the results:
>
> | MODEL       | SUN397   | Cars     | RESISC45 | EuroSAT  | SVHN     | GTSRB    | MNIST    | DTD      |
> | ----------- | -------- | -------- | -------- | -------- | -------- | -------- | -------- | -------- |
> | Pre-trained | 63.2     | 59.8     | 60.7     | 46.0     | 31.6     | 32.5     | 48.2     | 43.9     |
> | SUN397      | **75.0** | 47.0     | 54.3     | 46.5     | 28.3     | 26.4     | 44.3     | 41.6     |
> | Cars        | 56.6     | **78.3** | 50.9     | 38.4     | 30.2     | 30.6     | 49.7     | 41.8     |
> | RESISC45    | 52.0     | 47.2     | **95.2** | 56.9     | 23.9     | 24.3     | 39.7     | 35.9     |
> | EuroSAT     | 49.0     | 39.9     | 33.5     | **99.0** | 11.8     | 22.9     | 33.8     | 35.5     |
> | SVHN        | 40.5     | 36.3     | 18.9     | 9.8      | **97.3** | 27.3     | 81.8     | 23.2     |
> | GRSRB       | 36.9     | 33.0     | 20.6     | 21.3     | 41.2     | **98.9** | 30.9     | 23.9     |
> | MNIST       | 50.3     | 40.0     | 31.3     | 17.7     | 50.1     | 19.3     | **99.6** | 30.7     |
> | DTD         | 54.6     | 51.3     | 36.8     | 25.0     | 28.9     | 21.8     | 47.3     | **79.7** |
>
> **W2: A deeper explanation for the categorization of model ensembles, merging, and mixing would help clarify their task-specific benefits.**
>
> Model ensemble aggregates the predictions of multiple models to improve the overall performance, while all models in the ensemble are kept separate. Model merging combines the weights of multiple models into a single model, *keeping the model architecture the same*. Model mixing combines the weights of multiple models into a single model, *changing the model architecture*.
> For instance, with depth upscaling, a model mixing technique, the number of layers (depth) in the model differ from that of the original models.
>
> The pros and cons of each techniques:
>
> - Model ensemble: simple to implement, but expensive in terms of memory and computation resources for inference.
> - Model merging: no additional cost for inference, but the performance may not be as good as model ensemble.
> - Model mixing: can archive better performance than model merging, but it is more complex to implement. Typically requires additional training for performance recovery after fusion.
>
> **W3: More insights on scalability to additional tasks or modalities would enhance FusionBench’s applicability to diverse research needs.**
>
> That is a great suggestion. The benchmarking suite is designed to be easily extensible to new tasks and modalities, and we will include more tasks and modalities in the future.
>
> **W4: FusionBench’s effectiveness depends on high-quality pre-trained models, which might limit performance in under-resourced domains.**
>
> We agree that the performance of pre-trained models is crucial for the effectiveness of model fusion.
> In fact, the idea of model merging is widely used in machine learning to improve the performance of models with limited resources.
> For example, in each round of federated learning, the weights of the models from different clients are merged and distributed back to the clients. This can also be considered as a form of model merging.
> Other techniques for optimization such as exponential moving average (EMA), stochastic weight averaging (SWA), and latest weight averaging (LAWA) also use the idea of model merging to improve the performance and generalization of the models.

---

> > ### Author Response · Authors · 2024-11-14
> >
> > **Q1: Did you explore any non-fusion baseline models to provide a more direct comparison of fusion performance benefits? If so, could those results be included to clarify the relative impact of fusion methods?**
> >
> > Yes, we include the performance of task-specific models in the supplementary material and online documentation.
> >
> > **Q2:  Do you foresee expanding FusionBench to include multi-modal tasks, or would that require significant modifications to the existing benchmark structure?**
> >
> > Yes, we plan to include multi-modal tasks in the future.
> > In fact, many implemented algorithms are model-agnostic and task-agnostic, and can be applied to multi-modal models with minimal modifications.
> >
> > **Q3: Are there recommendations for tuning fusion methods based on specific task types, especially for users who may be new to model fusion?**
> >
> > Trying simple averaging or ensemble methods is a good starting point for users who are new to model fusion.
> > Simple averaging is a robust and architecture-agnostic method that can be applied to any model and domain, without the need for additional training.
> > After that, users can try to develop more advanced fusion method.
> >
> > **Q4: How does FusionBench handle variability in pre-trained model quality? Are there mechanisms to assess or adjust for model quality across tasks?**
> >
> > The pre-trained models indeed play a crucial role in the performance of model fusion, better pre-trained models will often lead to better performance of the fused model. In FusionBench, we provide a set of fine-tuned mdoels for each task to ensure the quality of models used in the fusion process are consistent across different fusion methods.

---

> > > ### Comment · Reviewer_qs86 · 2024-12-01
> > > **Official Response to Authors**
> > >
> > > The authors’ responses provide useful clarifications on certain aspects of the study and improve the understanding of FusionBench. The additional explanations and supplementary materials still lack novel foundational research in the field although I agree it can be of great value to the community. In my opinion, these responses fail to effectively establish its role as a breakthrough in foundational research.

---

### Author Response · Authors · 2024-11-14

We sincerely thank all the reviewers for their thorough evaluation and for offering insightful and constructive feedback. We are also pleased to see the positive recognition of our work's significance and practical implications, as well as the writing quality.

---

> ### Author Response · Authors · 2024-11-25
>
> Dear Reviewers,
>
> We sincerely appreciate you taking the time and effort to review our manuscript.  Your insights and feedback are precious to us.
>
> As the discussion phase nears its end, we want to confirm that we have adequately addressed all of your comments and concerns. We have carefully considered each point raised and believe we have provided comprehensive responses. However, if any questions remain unanswered or if there are areas where you feel further clarification or discussion is needed, please do not hesitate to let us know.  We are happy to provide additional information or engage in further discussion.
>
> Thank you again for your review and constructive feedback, which has significantly contributed to improving our work.
>
> Best regards,
>
> The Authors

---

### Comment · Area_Chair_gkgq · 2024-11-22
**Discussion**

Dear reviewers,

The authors have responded to your reviews.

Until November 26th @ 2359 (AOE time) reviewers and authors can freely exchange responses, so if there any clarifications you require from the authors, now is the time to seek them!

Best,

AC

---

### Meta-Review · Area_Chair_gkgq · 2024-12-13

**Metareview:**

This paper introduces FusionBench, which is designed to evaluate the performance of model fusion techniques across different tasks and domains. The reviewers were all appreciative of the engineering effort that went into making this work and the experiments conducted. There was criticism as to whether this was fundamental research, and that there were already existing tools that served a similar purpose. Several reviewer indicated that expanding the experiments would be beneficial (e.g. on large scale data, or multi-modal tasks).

The paper has review scores of 6,6,5,3 so is borderline. Reading the reviews, and looking at the paper, I feel that this is more a work-in-progress and isn't ready for acceptance yet, and would benefit from the additional experiments suggested by the reviewers; the authors themselves acknowledge the benefit of including multi-modal tasks in the future. There are a few insights gleaned (L469-479) but beyond this the research contributions are arguably quite small so I can see where Reviewer W4dN and qs86 are coming from with their criticisms.

**Additional Comments On Reviewer Discussion:**

There don't appear to have been any changes in score during the rebuttal period. After an exchange, Reviewer qs86 remained unconvinced about the research contributions of this work (but scored a 6). Reviewer ACwH did not reply the authors (although was prompted). Reviewer i5gE provided a short review, and acknowledged the author reply. Reviewer W4dN after an exchange remained unconvinced by research novelty.

I'm not inclined to be dismissive of benchmark work as "not being research" but there is a problem if it is not able to convince reviewers of its merit (which could be done with e.g. generalisable insights). This aside, there were a lot of comments effectively stating that there would be merit to additional experiments, which gave the impression that this is not yet a complete piece of work. It sounds like this paper would benefit from another iteration before acceptance.

---

### Decision · Program_Chairs · 2025-01-22

Reject